# Adjusting the Stiffness of Supports during Milling of a Large-Size Workpiece Using the Salp Swarm Algorithm

**DOI:** 10.3390/s22145099

**Published:** 2022-07-07

**Authors:** Krzysztof J. Kaliński, Marek A. Galewski, Natalia Stawicka-Morawska, Michał Mazur, Arkadiusz Parus

**Affiliations:** 1Faculty of Mechanical Engineering and Ship Technology, Gdańsk University of Technology, 80-233 Gdańsk, Poland; kkalinsk@pg.edu.pl (K.J.K.); margalew@pg.edu.pl (M.A.G.); natalia.morawska@pg.edu.pl (N.S.-M.); michal.mazur@pg.edu.pl (M.M.); 2Faculty of Mechanical Engineering and Mechatronics, West Pomeranian University of Technology in Szczecin, 70-310 Szczecin, Poland

**Keywords:** large-size workpiece machining, milling vibrations, stiffness adjustment, salp swarm algorithm

## Abstract

This paper concerns the problem of vibration reduction during milling. For this purpose, it is proposed that the standard supports of the workpiece be replaced with adjustable stiffness supports. This affects the modal parameters of the whole system, i.e., object and its supports, which is essential from the point of view of the relative tool–workpiece vibrations. To reduce the vibration level during milling, it is necessary to appropriately set the support stiffness coefficients, which are obtained from numerous milling process simulations. The simulations utilize the model of the workpiece with adjustable supports in the convention of a Finite Element Model (FEM) and a dynamic model of the milling process. The FEM parameters are tuned based on modal tests of the actual workpiece. For assessing simulation results, the proper indicator of vibration level must be selected, which is also discussed in the paper. However, simulating the milling process is time consuming and the total number of simulations needed to search the entire available range of support stiffness coefficients is large. To overcome this issue, the artificial intelligence salp swarm algorithm is used. Finally, for the best combination of stiffness coefficients, the vibration reduction is obtained and a significant reduction in search time for determining the support settings makes the approach proposed in the paper attractive from the point of view of practical applications.

## 1. Introduction

### 1.1. Problems in Milling of Large-Size Details

One of the main causes of the problems occurring during machining of large-size details are relative vibrations between the tool and the workpiece [1], which lead to deteriorated quality of the machined surface, increased tool wear, or even a destruction of the tool or the workpiece [2,3]. The problems of thermal dynamics, which may result in increased machining error or the occurrence of unfavorable residual stresses [4,5] are also important. They are characteristic for high-speed machining of hard-to-cut workpieces [6]. However, they play a smaller role in the issues of milling in the scope of moderate cutting speeds of large-size workpieces made of conventional materials (e.g., steel and cast iron). 

In manufacturing practice, in order to reduce the level of vibrations and avoid the abovementioned problems, usually “safe” parameters of machining are selected, for example, lower spindle rotation and feed speeds or small depth of cutting. This prevents undesired phenomena but, on the other hand, may limit machining process efficiency. There are various methods proposed to reduce, counteract, or circumvent vibrations during machining, developed mostly for small workpieces and in the context of chatter vibration reduction. However, it must be noted that there is an increasing demand for the production of large parts, and the current scientific research results fall behind the expected requirements in this field [7]. Additionally, due to the specifics of the milling process of the large-size workpiece, the vibration-related research should not be associated only with the chatter vibration phenomenon, which has also been noted, for example, in [7]. For the purposes of this paper, it was assumed that the category of the large-size workpiece includes items for which at least one linear dimension (i.e., length or width or height) is greater than 900 mm.

### 1.2. Vibration Reduction Methods

The developed vibration reduction methods may be briefly classified as out-of-process or in-process, and those that passively or actively modify the system’s behavior [1]. These methods, especially those dedicated to chatter reduction, are widely reviewed in [1,8,9,10].

For example, in [11], it was shown that different spindle speeds result in changes in the damping of system during turning, which results in different levels of stability limits. Moreover, active damping increases the limits of the system stability, especially in the low stability areas. In [12], the chatter vibration reduction solution during milling is presented, which is based on biaxial active actuator application designed by combining an inertial actuator and accelerometer working in a closed loop. The authors of [13,14] developed a noncontact electromagnetic actuator with two degrees of freedom integrated into the spindle system with differential driving mode to obtain a linear output of actuator force. The idea of active vibration reduction through active control of the spindle and tool position is also proposed in [15]. An intelligent tool holder is proposed in [16] and a tool with a built-in damper in [17]. A method of input shaping control technique to reduce vibrations in machining is proposed in [18], but its application is restricted by the dynamic properties of the feed drive, which, in the case of high-frequency vibrations, reduce operating performance. These methods are concentrated on reducing vibrations via active control of the cutting tool, tool holder, or the spindle with the tool. In [19], an active table is presented that enables vibration reduction owing to the movements of the whole workpiece mounted on top of the table. Many of these methods need sophisticated measurement equipment and actuators, which are not cost effective. Some types of actuators, for example, piezoelectric, are very sensitive to tensile or shear forces. Therefore, special design of actuator supports and housing are needed to allow its movement in the desired direction while simultaneously assuring high stiffness in other directions. Therefore, methods concerning active tables may be very difficult to apply in the case of large-size workpieces due to high mass and inertia of the workpiece and the table itself. 

Examples of semiactive methods include a tunable vibration absorber (TVA) [20] and a tuned mass damper (TMD) [21]. To determine the optimal spring stiffness and absorber position values of TVA, an optimal algorithm was developed based on the mode summation approach. In [21], it was shown that a two-DOF TMD (receiving translation and rotation motion) demonstrates better efficiency than single-DOF (SDOF) and two SDOF TMDs with equal mass. To minimize low frequency vibrations, a contactless multilayer electromagnetic spring with tunable negative stiffness is proposed in [22]. The electromagnetic force between the magnets and coils generates negative stiffness, which can be tuned online by controlling the current. In [23], the authors present an algorithm for tuning a semiactive clamping table for the purpose of chatter suppression in turbine blade recontouring. The damping of the table is provided by the adjustable eddy current damping modules. The use of electrorheological and magnetorheological fluids to simultaneously control the stiffness and damping of the vibration reduction devices has also been investigated and reported in [24]. 

The group of in-process solutions concerns, for example, various methods based on vibration suppression through spindle speed variation. In [25], a speed variation command is activated after detecting chatter occurrence. In [26], an adaptive speed modulation is proposed to suppress chatter, and in [27] an optimal–linear spindle speed control is proposed to avoid development of chatter vibrations. The drawback of these methods is the limited capability of spindle control systems to quickly change spindle speed and to perform this without stopping the feed. The other group of methods concerns matching the spindle speed to the selected dynamic properties of the cutting process, especially to optimal phase shift between inner and outer modulation [28], to the dominant natural frequencies of the vibrating system [29], or to minimize the vibration level of a workpiece [30]. These methods are easier to implement than all of the previous methods but may limit the overall efficiency of the milling process, for example, in the case when the spindle speed is optimal for vibration reduction but is much lower than the potential spindle speed that could be obtained by the milling center. 

Based on this review, the authors believe that there is still a need for development of methods that do not interfere with the milling machine structure, do not need complicated in-process monitoring, or do not require closed loop control, but, despite this, are capable of vibration reduction while maintaining milling process efficiency. This can be achieved, for example, by introducing the possibility for modifying elements of the dynamic properties of the tool–workpiece–support machine system.

### 1.3. Proposed Approach

From the point of view of vibration level during machining, the dynamic properties of the workpiece together with its supporting elements are essential [31,32,33] because they determine the workpiece’s susceptibility to vibrations. For example, properties such as resonant frequencies (if excited, chatter phenomena may easily occur) and damping coefficients (high damping prevents vibrations) are important. Although the properties of the detail itself cannot be modified, the parameters of the supports, for example, the stiffness, can be changed. This opens the opportunity to shape dynamic properties of the workpiece in a desired manner, for example, in order to reduce the level of vibration of the tool–workpiece during milling. For example, the authors of [34] propose the optimized sequence of tightening the anchor bolts for a given configuration of the machine bed and anchor system to ensure the correct stiffness of the machine tool foundation for machining large-size workpieces. Moreover, according to [35], surface quality during the face-milling process may be improved by optimizing the mounting pattern of the workpiece on the machine table base. This is possible because adjusting the support stiffness affects the modal parameters of the object, especially its dominant frequencies and modes of natural vibrations. This idea was also already proposed, for example, by the authors of [36,37] and is fundamental for the method proposed in this paper, as it consists of an application of special workpiece supports with adjustable stiffness. However, it must be noted that in previous works this concept was described only for small-size workpieces mounted in a single, variable stiffness holder. This is also a distinctly different approach than applied previously by authors, for example, in [27,30,38]. In the current paper, a different, original approach to solving the problem of reducing the vibration level of the tool–workpiece during milling is proposed. It consists in setting the stiffness coefficients of supports fastening the workpiece on the machine tool in such a way as to minimize the value of the previously defined vibration level indicator. To find the best set of support stiffness coefficient settings, a series of computer simulations of the milling process are performed. The simulations use a Finite Element Model (FEM) of the workpiece together with the supports and a dynamic model of the milling process. The parameters of the FEM are fine-tuned based on modal tests of the actual workpiece. Since the simulation of the milling process is time consuming, and the total number of simulations needed to search the entire available range of support stiffness coefficients is large, the modern and fast-converging Artificial Intelligence (AI) Salp Swarm Algorithm (SSA) [39] was used to significantly shorten the overall search time for the best combination of stiffness coefficients.

### 1.4. Metaheuristic Optimization Methods

The Salp Swarm Algorithm belongs to the wide and continuously growing group of metaheuristic AI algorithms. Metaheuristic algorithms can be categorized as evolutionary algorithms, human-based algorithms, swarm intelligence algorithms, and chemistry and physics algorithms [40]. Most of swarm algorithms are nature-inspired and usually they solve optimization problems by mimicking behaviors of various species of animals, for example, birds (in general) [41] or some particular bird species such as pigeons [42] or cuckoos [43], ants [44], grasshoppers [45], bees [46], bats [47], wolves [48], fish [49], krill [50], among many others. The main advantages of swarm optimizers are general simplicity, relatively easy implementation, and no information required for the objective function gradient. They are usually fast converging and can bypass local optima. Many are already applied in mechanical engineering problems [51], including milling [52,53] and turning operations [54,55].

### 1.5. Paper Organization and Research Program

The paper is organized as follows: first, the model of the milling process dynamics is described, which is used for simulations; then, the general procedure for adjusting the stiffness of supports is introduced and the salp swarm algorithm is presented, as this is the optimization method chosen to efficiently perform the search for the best set of stiffness coefficients, which is the main goal of the study. Next, the research and simulation example is presented that includes a description of the actual exemplary workpiece, its dynamic properties, and the properties of the adjustable stiffness supports. The selection of an appropriate indicator for comparing simulation results is also discussed. Then, the simulation results and application of the search procedure for finding the best set of stiffness coefficients are presented, and conclusions are drawn. 

## 2. Dynamics of the Face-Milling Process

The subject considered is the process of face milling of a flexible workpiece with a multiedge-milling cutter (Figure 1). The dynamics of the machining process were analyzed using a hybrid model, described in detail in [30,33,56], with the following assumptions:The tool fixed in the holder, rotating with the desired spindle speed *n*, and the workpiece mounted on the table, moving with the desired feed speed *υ_f_*, are the only features taken into account. The influence of the remaining parts of the milling machine on the dynamics of the machining process can be neglected [7,30].The flexibility of the workpiece, which characterizes the machining of large-size flexible elements on multiaxis machining centers, was taken into account [7,30].For modeling the dynamics of the cutting process, Coupling Elements (CEs) were adopted, which were located at the conventional contact points of the tool edges with the workpiece [56,57]. The momentary positions of the tips of the cooperating edges of the rotating tool were assumed as these points.Only cutting forces *F**_yl_*_1_, *F**_yl_*_2_, *F**_yl_*_3_, acting at the instantaneous point of contact of the selected tool edge with the workpiece (i.e., CE no. *l*), are taken into account. They work appropriately in the direction of cutting speed *v_c_*, cutting layer thickness *h_l_*, and layer width *b_l_*. Milling medium-cut materials (e.g., cast iron) with small allowances (depth of cutting *a_p_* = 0.2 mm, feed per edge *f_z_* = 0.17 mm) causes that the cutting force components acting on one edge do not exceed 200 NThese cutting forces depend proportionally on the instantaneous thickness of the cutting layer *h_l_* and the instantaneous width of the cutting layer *b_l_* [56,58]. Cutting speed *v_c_* values not exceeding 500 m/min allow the use of a mechanistic proportional model in the description of cutting dynamics [56].The passage of the current edge along the cutting layer causes a proportional feedback, and the passage of the previous edge additionally causes a delayed feedback. Because of this, it is possible to consider the effect of multiple trace regeneration in the calculation model.

When considering a face-milling process where a variable is observed in time, the $h_{l}\left(t\right)$ thickness and the $b_{l}\left(t\right)$ width of the cutting layer of the tool cutting edge no. *l*, we obtained [30]:(1)$$h_{l}\left(t\right)=h_{Dl}\left(t\right)+\Delta h_{l}\left(t-\tau_{l}\right)-\Delta h_{l}\left(t\right),$$
(2)$$b_{l}\left(t\right)=b_{D}-\Delta b_{l}\left(t\right),$$
where $h_{l}\left(t\right)$—cutting layer thickness at time-instant *t*; $h_{Dl}\left(t\right)$—nominal cutting layer thickness at time-instant *t*; $h_{Dl}\left(t\right)=f_{z}\sin\kappa_{r}\cos\varphi_{l}\left(t\right)$ for $f_{z}\ll D$ [38]; $\Delta h_{l}\left(\;\right)$—dynamic change in cutting layer thickness; $b_{l}\left(t\right)$—cutting layer width at time-instant *t*; $b_{D}$—nominal cutting layer width; $b_{D}=\frac{a_{p}}{\sin\kappa_{r}}$ [38]; Δb(t)—dynamic change in cutting layer width at time-instant *t*; $\tau_{l}$—time between the same position of edge no. *l*-1 and edge no. *l*; $\varphi_{l}\left(t\right)$—immersion angle of edge no. *l*; and *D*—pitch diameter of the cutting tool edges.

As a result of modeling the dynamics of the milling process, a system was obtained, consisting of (Figure 1): A structural subsystem, i.e., a rigid body called the Rigid Finite Element (RFE) no. *r* (central principal axes of inertia are *x_r_*_1_, *x_r_*_2_, *x_r_*_3_), that represents a milling tool connected to a tool holder by means of the Elastic Damping Element (EDE) no. *k*_1_ [56,57]. Its behavior is described in a domain of six generalized coordinates **q**.A modal subsystem, i.e., a stationary Finite Element Model (FEM) of a flexible workpiece supported by a finite number of Elastic Damping Elements (EDEs) no *k*_2_. At first, the subsystem is idealized as a set of tetrahedronal 10-node Finite Elements (FEs). The model obtained has a large number of degrees of freedom. Thus, it has been transformed to *mod* modal coordinates, whose number is much smaller [30].A connecting subsystem, i.e., a set of Coupling Elements (CEs), the positions of which correspond to the instantaneous positions of the tips of the tool edges and change with respect to time [30,56].

The momentary position of the cutter edge no. *l* is described by the immersion angle *φ**_l_* = *φ**_l_*(*t*). It corresponds to the temporary position of CE no. *l*, and the axes *y_l_*_1_, *y_l_*_2_, *y_l_*_3_ are the coupling axes of this CE [30,57]. During the machining process, not all edges are cutting the material at any given time. The cutting edges are called “active”.

In [30], it was shown that the equation of dynamics of the hybrid model obtained for the face-milling process has the form:(3)$$\left[\begin{array}{cc} M & 0\\ 0 & I \end{array}\right]\left\{\begin{array}{c} \ddot{q}\\ \ddot{a} \end{array}\right\}+\left[\begin{array}{cc} L & 0\\ 0 & 2Z_{c}\Omega_{c} \end{array}\right]\left\{\begin{array}{c} \dot{q}\\ \dot{a} \end{array}\right\}+\;\left[\begin{array}{cc} K+\overset{i_{l}}{\underset{l=1}{\sum }}T_{l}^{T}\left(t\right)D_{Pl}\left(t\right)T_{l}\left(t\right) & -\overset{i_{l}}{\underset{l=1}{\sum }}T_{l}^{T}\left(t\right)D_{Pl}\left(t\right){\tilde{W}}_{l}\left(t\right)\Psi_{c}\\ -\overset{i_{l}}{\underset{l=1}{\sum }}\Psi_{c}^{T}{\tilde{W}}_{l}^{T}\left(t\right)D_{Pl}\left(t\right)T_{l}\left(t\right) & \Omega_{c}^{2}+\overset{i_{l}}{\underset{l=1}{\sum }}\Psi_{c}^{T}{\tilde{W}}_{l}^{T}\left(t\right)D_{Pl}\left(t\right){\tilde{W}}_{l}\left(t\right)\Psi_{c} \end{array}\right]\left\{\begin{array}{c} q\\ a \end{array}\right\}=\;\left[\begin{array}{c} \overset{i_{l}}{\underset{l=1}{\sum }}T_{l}^{T}\left(t\right)F_{l}^{0}\left(t\right)+T_{l}^{T}\left(t\right)D_{Ol}\left(t\right)\Delta w_{l}\left(t-\tau_{l}\right)\\ -\overset{i_{l}}{\underset{l=1}{\sum }}\Psi_{c}^{T}{\tilde{W}}_{l}^{T}\left(t\right)F_{l}^{0}\left(t\right)-\Psi_{c}^{T}{\tilde{W}}_{l}^{T}\left(t\right)D_{Ol}\left(t\right)\Delta w_{l}\left(t-\tau_{l}\right) \end{array}\right],$$
where:

**M, L, K**—matrices of inertia, damping and stiffness of the structural subsystem; $\Omega_{c},\;\Psi_{c},\;Z_{c}\;$—matrices of angular natural frequencies, normal modes and dimensionless damping coefficients of the modal subsystem; **q**—vector of generalized coordinates of the structural subsystem; **a**—vector of modal coordinates of the modal subsystem; ${\tilde{W}}_{l}\left(t\right)$—transformation matrix between the generalized displacements vector of the modal subsystem and the displacements in the coordinate system *y*_l1_, *y*_l2_, *y*_l3_ of CE no. *l*; $T_{l}\left(t\right)$—transformation matrix of the generalized displacement vector of the structural subsystem **q** from the *x**_r_*_1_, *x**_r_*_2_, *x**_r_*_3_ coordinate system of RFE no. *r*, to the coordinate system *y_l_*_1_, *y_l_*_2_, *y_l_*_3_ of CE no. *l*; $F_{l}^{0}\left(t\right)$—vector of cutting forces of CE no. *l*, resulting from desired cutting geometry and kinematics; $D_{Pl}\left(t\right)$—matrix of proportional feedback interactions of CE no. *l*; $D_{Ol}\left(t\right)$—matrix of time-delayed feedback interactions of CE no. *l*; $\Delta w_{l}$—vector of deflections of CE no. *l*, $\tau_{l}$—time delay between the same position of CE no. *l* and of CE no. *l*-1, number of “active” CEs; the symbol *t* means the current moment of time, while *t* − *τ_l_*—the earlier moment, when the previous cutting tool edge was in the same geometrical place.

## 3. Adjusting the Stiffness of Supports—General Procedure and Search Method Selection

As described in the Introduction, in order to reduce vibration level during machining, mounting large-size details on a number of supports with adjustable stiffness is proposed (Figure 2).

In order to minimize the vibration level, an appropriate set of support settings must be determined. The general procedure for finding these settings is:Identification of stiffness values for different settings for each of the adjustable stiffness supports, for example, by performing static tests at the material testing machine;Preparing the modal model of the workpiece itself and assessing its compliance with the actual object using modal parameters identification methods (for example, ERA—Eigenvalue Realization Algorithm or p-LSCFD—poly-reference-Least Square Complex Frequency Domain methods [31,56,58]);Selecting cutting process parameters, i.e., depth of cutting, feed speed, spindle speed;Performing a series of simulations of milling process for given sets of support stiffness settings;Assessing the simulation results by comparing a chosen process quality indicator, for example, average tool–workpiece displacement or Root Mean Square (RMS) of the displacements in the time domain;Choosing the set of support settings that assure the best milling conditions.

As it can be seen, the procedure implies the performance of milling process simulations for different support settings. In the case of using a few supports for mounting the workpiece, and even limiting the number of possible settings for each support to a few values, the number of combinations becomes very large and the whole procedure may be very time consuming. This sets a challenge on how to efficiently search for the best set of support settings that assure vibration minimization. 

The solution to this problem is application of an effective optimization technique. A wide and still growing group of such algorithms are AI algorithms and, particularly, Swarm Intelligence algorithms. Swarm intelligence is defined as the collective behavior of decentralized, self-organized natural or artificial systems. Algorithms of this class are nature-inspired, metaheuristic algorithms that usually solve optimization problems by mimicking physical or biological phenomena, especially behaviors of various species of animals. In general, swarm optimization algorithms are relatively simple to implement, do not require information about the objective function gradient (contrary to many classic optimization algorithms), are usually fast converging, and can bypass local optima. The most notable feature of these methods is that they search for the optimum by moving individual search agents in the search space. There is also no centralized controller or supervisor of the whole swarm. Each agent follows usually quite simple rules and can perform elementary operations. Although one agent is unable to solve the problem alone, owing to the mutual interactions with other agents and with the environment (problem being solved), the whole swarm is able to “intelligently” find the solution. An example of successful application of the swarm intelligence method for Finite Element Model (FEM) updating, which is based on the Particle Swarm Optimization algorithm used for (non-adjustable) support stiffness coefficient estimation, is presented in [31].

Amongst swarm intelligence algorithms, the recent Salp Swarm Algorithm (SSA) [59] offers many advantages that make it appealing for solving the problem of efficient search for the best set of support stiffness. Although the SSA is a relatively new algorithm (published in 2017), it has already gained recognition because of its simplicity and properties. Some applications, mostly related to energy distribution and production systems, include, for example, optimization of wind turbine location [59], optimization of power system operations [60], estimation of the parameters of photovoltaic panels [61], and prediction of wind power [62]. Some other examples include UAV path planning [63], design of PID-fuzzy control against an earthquake for a seismic-exited structural system [64], and prediction of pressure burst in pipelines [65]. In all these tasks SSA showed very good performance, efficiency, and competitiveness, outperforming other well-established approaches [39,66]. According to [67], the main advantages of SSA are good convergence acceleration, efficient global performance and excellent solutions, suitability for many optimization tasks, good handling of wide search space, adaptability, robustness, scalability, and reasonable execution time. The only important disadvantages noted are possible premature convergence and probability distribution change occurring between algorithm iterations. Because of these advantages, SSA was selected as an algorithm that can help solve the problem of efficiently implementing the search for the best set of support stiffness coefficients.

## 4. Salp Swarm Algorithm

Salps are small, barrel-shaped, gelatinous organisms that move by pumping water through their bodies. They are common in oceans around the world. Salps may live alone but often form long, stringy colonies. As necessary for every living creature, salps need to search for food. These two behaviors, i.e., food chasing and swarming in chainlike forms, were inspiration for the Salp Swarm Algorithm which is a metaheuristic AI algorithm [39].

In the algorithm, a salp represents an individual search agent. During initialization, agents are randomly placed in the search space and one of them is selected as a chain leader, which moves towards the food, i.e., best solution found so far (Equation (4)): (4)$$x_{i}^{1}=\{\begin{array}{c} F_{current\;best}+c_{1}\left(\left(ub-lb\right)c_{2}+lb\right)\;\mathrm{for}\;c_{3}\geq 0.5\\ F_{current\;best}-c_{1}\left(\left(ub-lb\right)c_{2}+lb\right)\;\mathrm{for}\;c_{3}<0.5 \end{array}$$
where:

***x***—the vector describing the agent’s position; *i*—the current iteration number; 1 in the upper index in $x_{i}^{1}$ denotes the first (leader) salp; *ub*—the upper bound; *lb*—the lower bound of the search space.

According to (4), the leader position is updated only with respect to the best solution found so far (*F_current_*). However, movement of this agent (the leader) is distorted by *c*_1_, *c*_2_ and *c*_3_ coefficients. Coefficients *c*_2_ and *c*_3_ are random values with uniform distribution in the range <0, 1>. The *c*_1_ factor is calculated in each iteration as:(5)$$c_{1}=2e^{-{\left(\frac{4i}{I}\right)}^{2}},$$
where *i* is the current iteration number and *I* is the maximum number of iterations.

The *c*_1_ coefficient is a very important parameter of SSA as it balances exploration and exploitation. In the initial iterations, the value of *c*_1_ is close to 2 so the part of Equation (5) in parentheses dominates, but as the algorithm progresses, its influence is reduced and the leader starts to move in gradually smaller steps around the best solution found so far by the whole swarm. The *c*_1_ coefficient is a very important parameter in this algorithm because it weights its exploration and exploitation behaviors. It is also a practical advantage of SSA that there is only one adaptively decreasing hyperparameter (namely *c*_1_) that is calculated using Formula (6) and predefined by SSA authors so there is no need to search for its optimal value, as in the case of many other swarm intelligence algorithms. Apart the leader, all other agents move in the direction of the proceeding salp (for example, the one with the lower index on the list of agents) according to:(6)$$x_{i}^{j}=\frac{1}{2}\left(x_{i}^{j}+x_{i}^{j-1}\right)\;\mathrm{for}\;j\geq 2,$$
where *j* is the salp number.

This imitates chain-forming and leader-following behaviors (Figure 3). In the consecutive iterations, the leader is moved randomly around the current best solution (4) in a gradually decreasing range (due to (5)) while other salps, by following their predecessors, progressively shrink the chain (6). This is symbolically marked by arrows in Figure 3. Such agent position update rules help in refining the solution and, additionally, help the algorithm escape from local optima and prevent premature convergence.

## 5. Research and Simulation Example

### 5.1. Introduction

The procedure presented in the section entitled “Adjusting the Stiffness of Supports—Justification and General Procedure” is explained based on the practical example of the workpiece presented in Figure 4a. The workpiece main body size is approximately 1000 × 550 × 260 mm. It is made of EN-GJS-400-15 cast iron and weights approximately 175 kg. More details on the workpiece dynamic properties and on the characteristics of its supports are described in the subsequent sections.

### 5.2. Support Characteristics

Following the procedure described in the section “Adjusting the Stiffness of Supports—The General Procedure and Search Method Selection”, the values of the stiffness coefficients for the various settings of the adjustable stiffness supports must be identified first. Using a ZwickRoell Z020 testing machine, the values of the stiffness coefficients were obtained for the three supports: S1, S2, and S3. For each of the supports and for their five different settings (from 20 mm—the stiffest support, through 30, 40, 50 mm, to 60 mm—the least stiff support), the values of the stiffness coefficients were determined. Then, the linear approximation with the use of the determined values was performed and the static characteristics of the supports, presented in Table 1, were obtained. These approximate characteristics were used during the search for the best set of settings that involves simulations of the milling process. During the simulations, the model of the workpiece placed on the adjustable supports was used.

### 5.3. Workpiece Model

The schematics of the workpiece and its Finite Element Model (FEM) are presented in Figure 4. The FEM of the free-body workpiece was tuned according to the results of the modal test using the approach described in [31]. During the modal tests, the object was excited using The Modal Shop 2100E shaker (maximum force 440 N) (The Modal Shop Inc., Cincinnati, OH, USA), generating a burst random signal. The applied force was measured using a PCB 208C02 force sensor (±440 N) (PCB Piezotronics Inc., Depew, NY, USA). Object responses were measured using 16 DJB A/120/V accelerometers (±75 g) (DJB Instruments, Suffolk, UK). Acceleration signals were acquired via NI PXI-4496 and force signal via NI PXI-6221 DAQ cards (National Instruments Corp., Austin, TX, USA).

In [31], the relationship between the angular natural frequencies $\Omega_{f}$ and normal modes $\Psi_{f}$ of free-body workpiece vibrations, and the corresponding parameters $\Omega_{c}$ and $\Psi_{c}$ of the modal model of the workpiece mounted on the machine table with the use of *i**_p_* rigid supports, which constitute additional constraints, was demonstrated. It was assumed that the modal coordinates in the number of *mod* take into account:All rigid body modes of the *free–free* state system;A set of the first few elastic modes of the *free–free* state system, the number of which should be not lower than the number of flexible modes of the constrained system to be computed;All Guyan modes [68] of a *free–free* state system for all combinations of Degrees of Freedom (DOF) of the supports.

As a result, the matrix $\Psi_{c}$ of *mod* normal modes of natural vibrations of a coupled (constrained) system is described by the formula:(7)$$\Psi_{c}=\Psi_{f}T_{\Psi },$$
and the matrix of angular frequencies of natural vibrations of the coupled system $\Omega_{c}$ results from the dependence:(8)$$\Omega_{c}^{2}=T_{\Psi }^{T}K_{\Psi }T_{\Psi },$$
where:

$T_{\Psi }$—matrix of eigenvectors of the following matrix:(9)$$K_{\Psi }=\Omega_{f}^{2}+\Psi_{f}^{T}W^{T}K_{s}W\Psi_{f};$$$K_{s}$—the stiffness matrix of the free system supports, which are constraints; W—modal coupling matrix, whose values for the constrained DOF (connected to the support) should be 1 and 0 otherwise.

The computed angular frequencies of natural vibrations $\omega_{c,i}^{a},i=1,\ldots ,\;mod$ can be directly compared with corresponding angular frequencies $\omega_{c,i}^{e},i=1,\ldots ,\;mod$ measured during the modal tests, obtained from the experiments. To compare the *i*th computed vector $\Psi_{c,i}^{a}$ with the *j*th experimental vector $\Psi_{c,j}^{e}$ of normal modes, the Modal Assurance Criterion (MAC) can be used [69]:(10)$$MAC\left(\Psi_{c,i}^{a},\;\Psi_{c,j}^{e}\right)=\frac{{\left(\Psi_{c,i}^{aT}\cdot \Psi_{c,j}^{e}\right)}^{2}}{\left(\Psi_{c,i}^{aT}\cdot \Psi_{c,i}^{a}\right)\cdot \left(\Psi_{c,j}^{eT}\cdot \Psi_{c,j}^{e}\right)}.$$

MAC values close to 1 for the normal modes indicate a linear dependence of mode shapes. Even if the MAC factor is generally considered not to be a perfect tool for measuring the correlation between mode shapes [69], it was used for fast correlation assurance and was assumed sufficient.

Owing to the modal tests, the six vibration modes of the workpiece were identified using the p-LSCFD method [31,56]. The FEM model presented in Figure 4b was appropriately fixed with *i**_p_* = 12 finite elements representing springs (each with six degrees of freedom) to obtain satisfactory correlation of natural frequencies and modes [31]. Appropriate values of the measured and calculated frequencies of the natural vibrations of the workpiece mounted on the milling machine are compared in Table 2. The values of the Modal Assurance Criterion (MAC) are presented in Table 3. 

The validation of the computational model of the workpiece mounted on the milling machine allowed, first of all, to identify the parameters of the modal model of the free-body workpiece, i.e., the angular natural frequencies $\Omega_{f}$ and normal modes $\Psi_{f}$ matrix. 

This prepared model of the workpiece was used during simulations of the milling process that were performed for various settings of the adjustable supports.

### 5.4. Milling Process Simulations

First, the simulation of the face milling of a surface indicated in Figure 4a for a workpiece fixed with three non-adjustable, high stiffness supports was performed using: ∅63 mm cutter with six edges, tool rotation speed *n* = 1112 rev/min, feed speed *υ_f_* = 1112 mm/min, and depth of cutting *a**_p_* = 0.2 mm. These parameters were selected according to the standard parameters used during milling of the actual workpiece by the cooperating industry (PHS Hydrotor Inc., Tuchola, Poland). The result of this simulation is presented in Figure 5. The meaning of the vibration level indicators presented in the figure is described in Section 5.5.

During the search for the best set of adjustable support settings, clamping elements were removed from the computational model of the workpiece. Then, Elastic Damping Elements (EDEs) with known stiffness coefficients, which are computational models of three supports with adjustable stiffness, were added at specific points to the modal model of the free-body workpiece obtained in the way presented above, with the properties described by the matrices $\Omega_{f}$ and $\Psi_{f}$**,**. The stiffness coefficients of all the remaining supports in the model were set to zero. Hence, the new stiffness matrix $K_{s}$ of these EDEs was known. Based on the dependencies (7) and (8), the matrices $\Omega_{c}$ and $\Psi_{c}$ were again determined for the modal model of the workpiece bound by the constraints. 

Milling simulations based on the abovementioned computation model (3) were performed using proprietary software. To assess the vibration level for a given set of support settings, one simulation run was needed. For the workpiece considered in this paper, from 20 to 65 s were required, depending on the type of computer used for the simulations. (Intel Core i7 10th and 11th generation and Core i5 3rd, 4th, and 8th generation were used). This means that the search for the best set of settings, which requires performing a high number of simulations, may be very time consuming.

### 5.5. Selection of an Indicator for Comparing Simulation Results

In order to select the set of adjustable stiffness support settings, it was necessary to choose an indicator that would allow for comparisons between individual simulation results. Various indicators were considered: ***Av***—average (mean) value of the displacements—takes into account mainly “static” displacement of the workpiece. The amplitude of vibration around the average value is not represented by this indicator; however, it may be associated with the accuracy of geometrical requirements for the workpiece.***RMS***—Root Mean Square value of the displacements—the general indicator of vibrations; however, it is strongly influenced by the mean value if it is non-zero.***RMS_Av_***—Root Mean Square value of the displacements but calculated for the signal after subtracting its mean value. This indicator is mathematically equivalent to the standard deviation of the observed displacements. It does not take into consideration the mean value. The advantage of this indicator is that it represents relative tool–workpiece vibrations (and not “static” displacement of the workpiece). Thus, it may be related to machined-surface quality.

To assess characteristics of each of these indicators, a set of 729 simulations were performed. The number of simulations result directly from the assumed number of nine different settings for each support (20, 25, 30, 35, …, 60 mm). Figure 6 presents the consolidated results for the *Av* and *RMS**_Av_*** indicators. For each indicator, calculations were performed only for the time period of the machining when there were no transient processes (i.e., tool entry and exit at the beginning and at the finish of milling). An example of one simulation result is presented in Figure 7. The yellow line presents the *Av* indicator value. The line is drawn only for the time range during which the indicator was calculated.

After the performance of preliminary simulations, it was observed that average and *RMS**_Av_*** indicators tend to show the opposite location of the best set of support settings, the stiffest setting is preferred by the first one and less stiff regions by the other, as they concentrate only on the one aspect of milling results (“geometrical” accuracy vs. relative vibrations level) (compare Figure 6a,b). This led to the conclusion that probably the most appropriate solution was to develop an indicator that accounts for both “geometrical” accuracy and relative vibrations. According to this, the indicator *F = Average + RMS**_Av_*** was proposed. However, further comparison of the results showed that the lowest observed value for *Average* was 0.002026 mm (for S1 = 20 mm, S2 = 20 mm, S3 = 20 mm), and the lowest value for *RMS**_Av_*** is 0.000656 mm (for S1 = 45 mm, S2 = 55 mm, S3 = 55 mm) (red points in Figure 6a,b). This means that *RMS**_Av_*** has roughly one third the influence of the *Average* on the total *F* value. In order to balance this, an indicator: (11)$$F_{w}=Average/weight+RMS_{Av}\;$$
with *weight* = 3 was selected. This *weight* value was chosen as the most appropriate one to assess results of simulations during search for the best set of support settings. The consolidated result of simulations for the *F*_1_ and *F*_3_ indicator are presented in Figure 8a,b. The more modest impact of average vibration level on the overall indicator value can be noticed. More uniform colors mean that indicator values are more leveled in almost the whole range of settings. This confirms that the *F*_3_ indicator is well balanced. A particular example of simulation results and differences in *A_v_*, *F*_1_ and *F*_3_ values is also presented in Figure 7.

### 5.6. Search for the Best Set of Settings Using the Salp Swarm Algorithm

As mentioned earlier, the preliminary simulations were time consuming: 9^3^ = 729 individual simulations were performed. Despite such a large number of simulations, it did not guarantee that the best set of support settings obtained was the best in the global scale, because it may lie between the settings considered in this set. Choosing a higher mesh density for the support settings would dramatically increase the number of simulations. For example, a change from 5 to 2 mm would raise this number to 21^3^ = 9261 and extend the overall simulation time almost 13 times. It became obvious that an effective optimization technique was needed to obtain the best solution in a reasonable time. Because of the previously mentioned advantages, the SSA was chosen to perform this task.

## 6. Results and Discussion

Several support setting selection runs were performed using SSA. The goal was to evaluate if it is possible to obtain solutions similar to the solution known from the preliminary simulations (or better) in much shorter time (i.e., with the lower number of individual simulations). Various combinations of the number of agents and search steps were considered. For each combination, five SSA passes were performed to limit the potential randomness of the results due to random initial positions of the agents and random values of *c*_1_, *c*_2_, and *c*_3_ coefficients in the algorithm. The search space was limited to <20 … 60> mm for each dimension, which covers all of the possible settings for each support. Table 4 presents the summary of the results. For comparison, the result for a deterministic search for the 5 mm step of support settings, and its subset for the 10 mm step (<20, 30, …, 60> mm) are included in the table. The latter set shows the same results as for the 5 mm step, only by coincidence, because the best solution was found for the setting: S1 = 20 mm, S2 = 20 mm, and S3 = 50 mm. In general, a lower resolution of the search space reduces the chance of finding the actual best solution, as it may lie between the values considered.

As can be observed in Table 4, in all of the cases where SSA was applied, the best solution found (for a given set of passes) is very close to the best solution found during all of the tests (less than 1% of error—column 6 in Table 4), and the average error for each set is below 2.5% (column 8). Even for the worst case, the error was below 5.5% (column 7). Additionally, the best result found is better than the result of preliminary simulations for fixed support settings steps. Figure 9 presents one selected SSA pass which resulted in the best solution according to the *F*_3_ indicator. The characteristic sight, which is a visual effect of the shrinking salp chain, may be observed. Figure 10 presents the progress of this pass, i.e., the lowest value of the *F*_3_ indicator found so far and an average value of this indicator for all of the agents in each step. This is a typical convergence curve that can be observed with various optimization algorithms. Results of simulations of relative tool–workpiece displacements for the best set of support settings, obtained from the SSA, is presented in Figure 11. It can be compared with Figure 7, which was the best result obtained for the *Av* indicator in the deterministic search. Lower levels of vibrations, not only in terms of the *F*_3_ indicator values but also in the vibration maxima and general signal envelope, can be observed.

According to Table 4, the results from 200 SSA simulation passes have very low average error, below 1%, and below 2.5% for 120 simulation passes. Running a pass more than once helps to obtain better results and assures consistency in the results. For example, in most cases, after five passes, the best solution error dropped below 0.6% (column 6). Of course, performing multiple passes raises the total number of simulations, but even for five passes with 120 simulations, the total number of simulations is still lower than for a deterministic, blind-search of 729 setting combinations.

To bring additional context to the quality of the results, it must be noted that the highest value of the *F*_3_ indicator was 0.476 mm, which is 328 times higher than the minimal value obtained, *F*_3_ = 0.00144803 mm. However, this value was calculated for a case where, due to an adverse combination of support settings, vibrations were very high (see the brightest dots in Figure 8b). To limit the considered range of *F*_3_ to reasonable values, these border cases should be omitted. If we reject 3% of such cases (i.e., 22 out of 729 from preliminary simulations), then the maximum becomes *F*_3_ = 0.00461 mm, which is 318% of the minimum value presented above. This underlines that the results obtained were very close to the best possible in all of the presented cases, and even the worst result was not far from it. 

Additionally, a close analysis of all of the simulations results revealed that in the presented case the search space contains many local extrema and, at the same time, regions where the *F*_w_ indicator sensitivity to support parameter changes is low. For example, small changes in the indicator value can be observed for the setting S1 ≈ 20 mm, S2 ≈ 20 mm, S3 = <45…55> mm. Despite this, in all of the passes SSA provided results close to the best one, with error below 5.5%.

The best set of support settings is S1 = 20 mm, S2 = 20 mm, S3 = 50.59 mm. For this case, indicator values are as follows: *Av* = 0.002073 mm, *RMS* = 0.002207 mm, *RMS_Av_* = 0.000757 mm, *F*_1_ = 0.00283 mm, and *F*_3_ = 0.001448 mm. It must be noted that this was obtained on the basis of minimization of the *F*_3_ indicator only, so all of the other indicator values may not be optimal. However, all of the results, both for the deterministic set of simulations and for the SSA search, showed that a proper setting of the support stiffness may result in vibration reduction. Especially, for the best case *RMS_Av_* was reduced by about 50% and *F*_3_ by about 10%.

## 7. Conclusions

The procedure described in the paper consists of the following steps: a modal test of the real workpiece; identification of the modal model of the workpiece for the purpose of milling process simulations; modification of the workpiece; development of a FEM model by replacing non-adjustable supports with adjustable stiffness supports; and, finally, searching for the best set of settings of these supports by performing a series of milling simulations for various support settings.

Owing to the results presented, it can be concluded that:Introducing adjustable stiffness supports for the workpiece may result in a reduction in vibration that occurs during the milling process. For example, the *RMS_Av_* value, which may represent relative tool–workpiece vibrations that influence machined-surface quality, was reduced by around 50%. However, efficient determination of the set of appropriate settings may be challenging, especially due to the need for performing a large number of time-consuming simulations. To overcome this issue, the salp swarm algorithm was applied.The salp swarm algorithm is a very efficient, fast-converging algorithm that deals very well with multidimensional problems having a complex search space. This is important for solving problems where the number of algorithm steps must be limited, for example, in cases when each step involves time-consuming calculations or simulations, which was the case presented in this paper.Application of the SSA algorithm leads to a significant time reduction needed for searching the space of decision variables, in terms of searching for the best combination of the stiffness coefficients for supports fastening a large-size workpiece on a milling machine. This makes the approach proposed in the paper attractive from the point of view of practical applications in manufacturing companies, where the problem of meeting the condition of minimizing the vibration level of the tool–workpiece has a significant technical importance (meeting the requirements for geometrical accuracy and quality of the processed surface) and economic importance (reduction in production time and required financial outlays).

In the perspective of further research, the most urgent planned task is to empirically confirm the correctness of the simulated results of vibration reduction in face-milling processes under industrial conditions. Then, the presented method can be extended to online control of the stiffness of the supports. Thus, the focus would be on redesigning the existing workpiece fixation supports, allowing the fixture stiffness coefficients to be continuously adjusted during machining. Another promising, and probably easier to implement, direction of research and development is a search for ways to reduce vibrations of the tool–workpiece during large-size machining with the use of control in a semiactive system, based on vibration dampers with variable parameters.

## Figures and Tables

**Figure 1 sensors-22-05099-f001:**
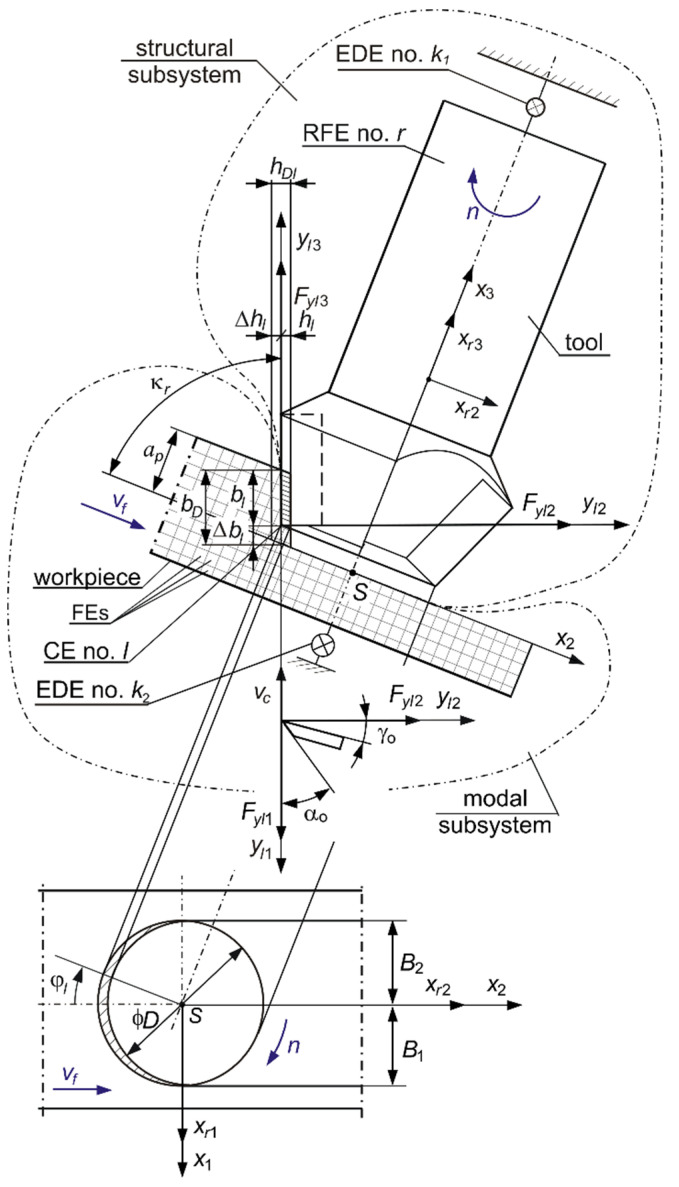
Scheme of the face-milling process of the flexible surface of the workpiece showing the modal subsystem created on the basis of the validated FEM of the workpiece. *κ_r_—*edge angle, *γ*_0_—edge rake angle, and *α*_0_—edge clearance angle.

**Figure 2 sensors-22-05099-f002:**
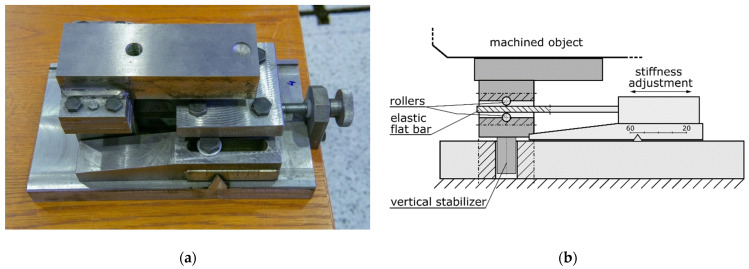
Workpiece support with adjustable stiffness coefficient: (**a**) general view, (**b**) schematics.

**Figure 3 sensors-22-05099-f003:**
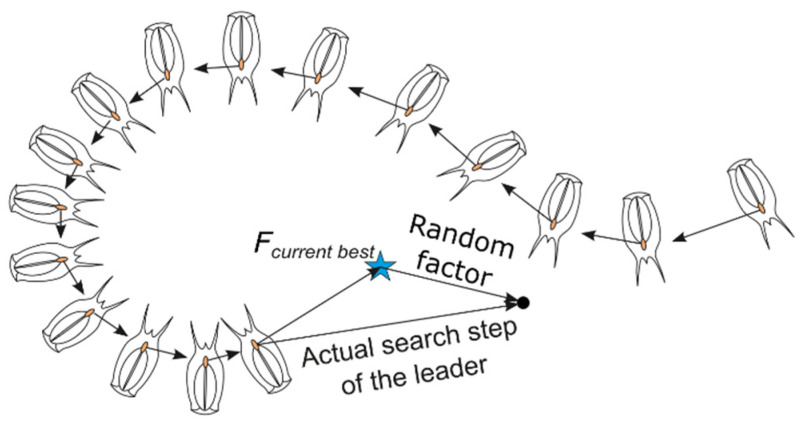
Salp swarm: chain-forming and food-chasing schematic.

**Figure 4 sensors-22-05099-f004:**
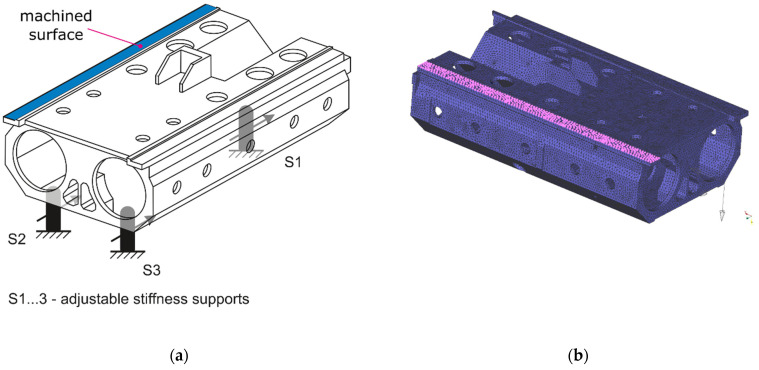
(**a**) Schematic of the workpiece and placement of the supports; (**b**) FEM of the workpiece.

**Figure 5 sensors-22-05099-f005:**
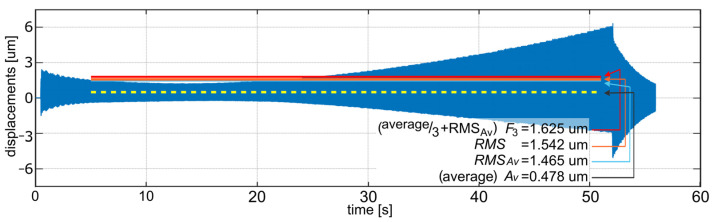
Milling process simulation results (relative tool–workpiece displacements) for 3 non-adjustable supports.

**Figure 6 sensors-22-05099-f006:**
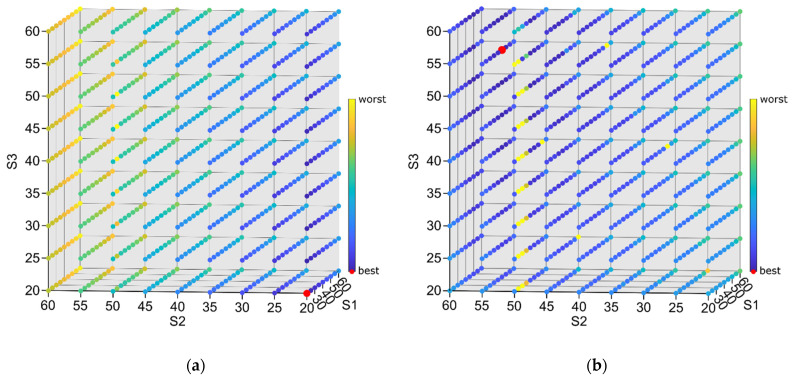
Consolidated simulation results for the (**a**) *Average* and (**b**) *RMS**_Av_*** indicators; darker dots represent better (lower) indicator values, and red marks represent the best (lowest) result.

**Figure 7 sensors-22-05099-f007:**
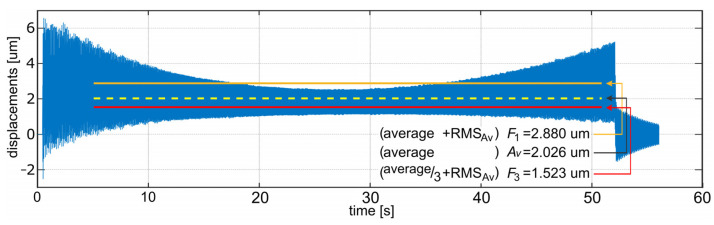
Example of simulation results (relative tool–workpiece displacements) for support settings: S1 = 20 mm, S2 = 20 mm, S3 = 20 mm.

**Figure 8 sensors-22-05099-f008:**
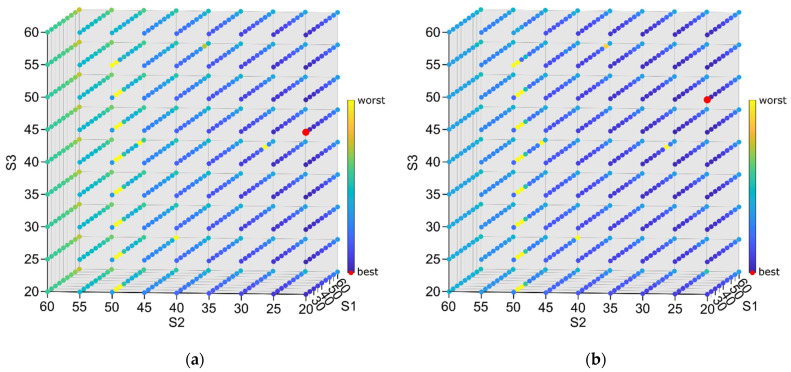
Consolidated simulation results for the (**a**) *F_1_ = Average + RMS_Av_* and (**b***) F_3_ = Average/3 + RMS_Av_* indicator; darker dots represent better (lower) indicator values, and red marks represent the best (lowest) result.

**Figure 9 sensors-22-05099-f009:**
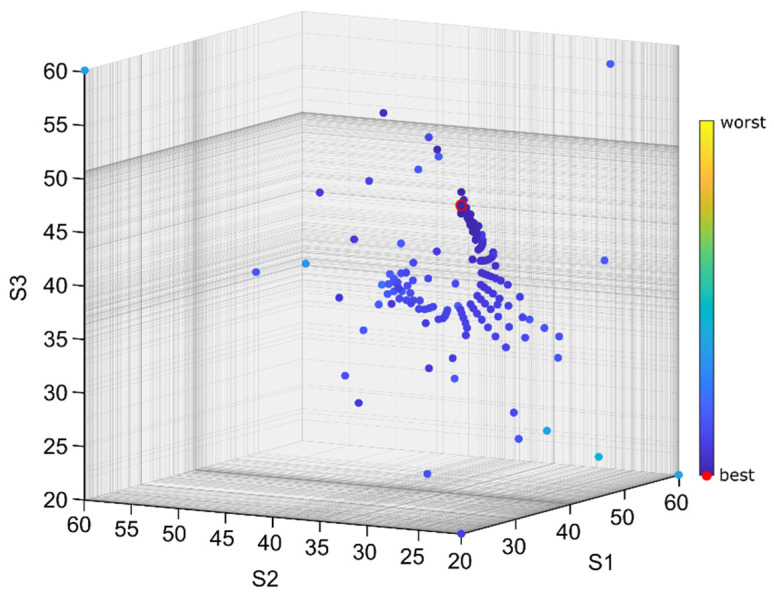
SSA pass resulting in the best solution according to the *F*_3_ indicator.

**Figure 10 sensors-22-05099-f010:**
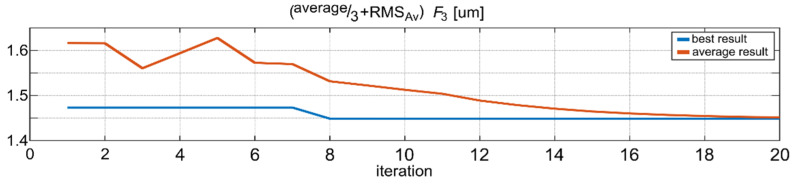
Progress of SSA pass resulting in the best solution according to the *F*_3_ indicator.

**Figure 11 sensors-22-05099-f011:**
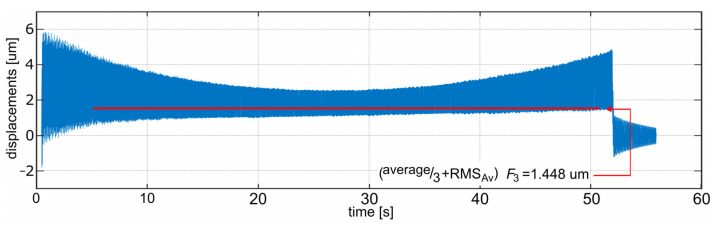
Simulation results (relative tool–workpiece displacements) for support setting: S1 = 20 mm, S2 = 20 mm, S3 = 50.59 mm.

**Table 1 sensors-22-05099-t001:** Stiffness coefficients *k* of supports with adjustable stiffness for different support settings *s*.

Support Number	*s* [mm]					Linear Approximation Characteristics
	20	30	40	50	60	
	*k* [N/mm]					
S1	15,515.24	13,353.17	10,492.44	7981.53	6218.45	*k* = −239.65 *s* + 20298
S2	14,678.93	12,685.56	9879.97	8113.67	6030.11	*k* = −218.70 *s* + 19025
S3	9669.46	8050.94	7064.00	5573.94	4683.09	*k* = −124.50 *s* + 11988

**Table 2 sensors-22-05099-t002:** Comparison of the measured and calculated frequencies of the natural vibrations of the workpiece mounted on the milling machine.

ωc,ie2π [Hz]	185.0	211.3	242.4	-	-	435.7	585.0	631.2
ωc,ia2π [Hz]	184.6	211.4	242.2	295.5	434.3	434.4	571.5	630.2

**Table 3 sensors-22-05099-t003:** Modal Assurance Criterion (MAC) values between FEM modes ($\Psi_{c,i}^{a}$) and modes identified during the modal test ($\Psi_{c,j}^{e}$), the highest values in bold.

	$\Psi_{c,1}^{e}$	$\Psi_{c,2}^{e}$	$\Psi_{c,3}^{e}$	$\Psi_{c,4}^{e}$	$\Psi_{c,5}^{e}$	$\Psi_{c,6}^{e}$	$\Psi_{c,7}^{e}$	$\Psi_{c,8}^{e}$
$\Psi_{c,1}^{a}$	**0.96**	0.05	0.08	-	-	0.01	0.09	0
$\Psi_{c,2}^{a}$	0.01	**0.98**	0.24	-	-	0	0.01	0.03
$\Psi_{c,3}^{a}$	0.08	0.18	**0.95**	-	-	0.03	0.05	0.01
$\Psi_{c,4}^{a}$	0.2	0	0	-	-	0	0.03	0
$\Psi_{c,5}^{a}$	0.13	0.08	0.03	-	-	0.01	0.18	0
$\Psi_{c,6}^{a}$	0.08	0	0	-	-	**0.94**	0.06	0.01
$\Psi_{c,7}^{a}$	0.05	0.03	0.03	-	-	0.05	**0.87**	0.01
$\Psi_{c,8}^{a}$	0	0.03	0	-	-	0.01	0.01	**0.94**

**Table 4 sensors-22-05099-t004:** Deterministic and SSA search results for the ***F*_3_** = *Average*/3+*RMS_Av_* indicator.

1. No. of Search Steps	2. No. of Agents	3. No. of Simulations	4. No. of Passes	5. Best Result [μm]	6. Best Result Error [%]	7. Worst Result Error [%]	8. Average Error [%]	9. Best Setting S1, S2, S3 [mm]
3 non-adjustable supports		1		1.62536				
Deterministic— 5 mm step		729	1	1.44857	0.04			20.00, 20.00, 50.00
Deterministic— 10 mm step		125	1	1.44857	0.04			20.00, 20.00, 50.00
9	13	117	5	1.44835	0.02	4.21	2.03	20.00, 20.00, 45.83
13	9	117	5	1.44832	0.02	1.27	0.86	20.26, 20.03, 51.47
8	15	120	5	1.45544	0.51	3.64	1.56	22.81, 21.84, 48.12
15	8	120	5	1.45606	0.55	3.38	1.85	23.34, 22.85, 46.81
7	18	126	5	1.44851	0.03	5.07	2.16	20.00, 20.00, 45.67
18	7	126	5	1.45387	0.40	5.35	2.46	23.38, 20.48, 44.66
10	20	200	5	1.45900	0.76	1.46	1.00	23.34, 20.30, 38.82
20	10	200	5	1.44803	0.00	1.42	0.60	20.00, 20.00, 50.59
	**Best**			1.44803	0.00	1.27	0.60	20.00, 20.00, 50.59
	**Average**			1.45218	0.29	3.23	1.57	

## Data Availability

The data presented in this study are available on reasonable request from the corresponding author.

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
