# Peer review of "Adjusting the Stiffness of Supports during Milling of a Large-Size Workpiece Using the Salp Swarm Algorithm"

_sensors, 2022, doi:10.3390/s22145099_

Round 1

Reviewer 1 Report

The authors employs Salp Swarm Algorithm to determine the best combination of stiffness coefficients for supports fastening a large-size workpiece during milling of flat surfaces. The change in the stiffness of the supports affects the modal parameters of the object, especially its dominant frequencies and modes of natural vibrations. Adjusting the supports’ stiffness can be achieved by using pre-marked brackets with adjustable mounting stiffness that replaces the standard rigid supports. In order to reduce the vibration level of the tool-workpiece during milling, it is recommended to set the supports’ stiffness coefficients in such a way as to minimize the value of the selected vibration indicator. The selection of this indicator is discussed in the paper. To find the best set of supports settings, a series of computer simulations of the milling process must be performed. The simulations use a model of the workpiece with supports in the convention of the Finite Element Model (FEM) and a dynamic model of the milling process. The FEM parameters are fine-tuned based on modal tests of the real workpiece. Since simulating the milling process is time consuming and the total number of simulations needed to search the entire available range of support stiffness coefficients is too large, the fast-converging modern, Artificial Intelligence Salp Swarm Algorithm is used to shorten the overall search time for the best combination of stiffness coefficients.

The paper is well-written and could be published after major revision.

More references from MDPI (sensors, metals, materials,….etc) should be added to the introduction section.

The role of temperature in machining process and its effects on workpiece integrity should be discussed, please see “temperature field sensing of a thin-wall component during machining: Numerical and experimental investigations”.

The introduction section could be supported by Fine-tuned artificial intelligence model using pigeon optimizer for prediction of residual stresses during turning of Inconel 718; A comprehensive review on residual stresses in turning; Prediction of residual stresses in turning of pure iron using artificial intelligence-based methods.

“Figure 1. Scheme of the face milling process of the flexible surface of the workpiece, modal subsystem created on the basis of the validation of the FEM of the workpiece.”; please add a reference for this figure.

Why did you use t and τ in equation (1); please use common symbols.

3D model or schematic drawing should be included in Figure 2.

Add a reference for Figure 3.

You should mention other metaheuristic models in your study, the limitations and advantages of your model should be compared with other optimizers, please see, Advanced metaheuristic techniques for mechanical design problems or A review on meta-heuristics methods for estimating parameters of solar cells.

What are S1, S2 and S3 in figure 4, explain them in the figure.

“To bring additional context to the quality of presented results it must be noted, that the highest value of F3 indicator was F3 = 0.476, which is 328 times higher than the minimal value.” What is the value of minimal value?

Conclusion section should be modified, use bullets.

Author Response

Dear Reviewer,

thank you for the valuable remarks.  Below you can find our reply with information about changes introduced in the paper.

Reviewer 2 Report

REVIEW

on article

Adjusting the stiffness of supports during milling of a large-size workpiece using the Salp Swarm Algorithm

Krzysztof J. Kaliński, Marek A. Galewski, Natalia Stawicka-Morawska, Michał Mazur and Arkadiusz Parus

SUMMARY

The article submitted for review is devoted to a hot topic, such as "Adjusting the stiffness of supports during milling of a large-size workpiece using the Salp Swarm Algorithm." The topic of the considered article is relevant. The paper presents the use of the Salp Swarm Algorithm to determine the best combination of stiffness coefficients for large workpiece fastening supports when milling flat surfaces. The research problem lies in the fact that the change in the rigidity of the supports affects the modal parameters of the object, especially its dominant frequencies and eigenmodes.

The authors note that the adjustment of the stiffness of the supports is achieved using pre-marked brackets with adjustable stiffness of fastening, replacing the standard rigid supports. The authors have done a lot of work, given useful recommendations, in particular, on setting the stiffness factors of the supports in such a way as to minimize the values of the selected vibration index, this is what the authors discuss in the article.

Thus, the authors used a modern methodological apparatus, the study is quite interesting, original and has a certain degree of scientific novelty and practical significance. At the same time, there are several shortcomings in the article that need to be corrected. They will be discussed below.

 COMMENTS

1.      The Abstract should contain a clearer formulation of the research problem. What problem did the authors solve and why is it necessary to do this, what is the rationale for the need for the study? This should be reflected in the Abstract. The authors must bring the Abstract in accordance with the journal’s requirements. Editors strongly encourage authors to use the following style of structured abstracts, but without headings: (1) Background: Place the question addressed in a broad context and highlight the purpose of the study; (2) Methods: Describe briefly the main methods or treatments applied; (3) Results: Summarize the article's main findings; and (4) Conclusions: Indicate the main conclusions or interpretations. The Abstract should be an objective representation of the article.

2.      In addition, the Abstract should clearly indicate the scientific novelty. If there are no questions from the point of view of practical significance, then from the point of view of scientific novelty, a clearer formulation should be given. What is the main scientific result of the author, what is new for modern science?

3.      The Introduction is too short and does not reflect a clear level of scientific novelty of the study.

4.      Too few analyzed sources are presented. For such a hot topic, you should increase them to at least 30.

5.      In addition, listing in one paragraph, in particular, on lines 46 to 53 of twenty sources, should be avoided. This is methodologically not entirely correct, more detailed information should be given, in which work were considered, what issues and why the authors see this scientific deficit after the review.

6.      A clear statement of scientific novelty, practical significance, goals and objectives of the study is required at the end of the Introduction section. Without this, it is impossible to proceed to conduct a direct study.

7.      The lack of detailed methods and materials used in the study is striking. The research program or statement of methods should be placed at the beginning of Section 2 or placed at the end of Section 1 of the Introduction.

8.      Figure 1 is quite cumbersome, difficult to understand and needs additional explanations. Figure 1 should be explained in more detail.

9.      The photo in Figure 2 should be presented in a higher quality.

10.    The statement on lines 126-127 contains references to 2 sources, but it looks very concise, so a slightly more extended interpretation of this statement should be provided.

11.    The transition between formulas 2 and 3 in section 4 should be given a little more broadly to make the narrative look more structured.

12.    Figure 3 looks interesting enough but needs more explanation. It is desirable to give a textual interpretation after the figure itself, then it will be more methodologically correct, because at present section 4 ends with a figure - this is not entirely correct.

13.    Figure 4b is presented in a rather low quality and needs to be improved.

14.    Section 5.1 is too short and concise; some explanation should be given and a smooth transition between subsections 5.1 and 5.2 should be made.

15.    Perhaps the design of table 3 should be revised due to the not entirely correct design of this table.

16.    Probably, the color scheme chosen in Figures 5a and 5b is not entirely successful. Perhaps this will look ponderous to the reader. The same remark applies to Figures 7a and 7b.

17.    Figures 8, 9 and 10 should be more detailed, as Section 6 Results looks very dry and lacks detail in particular? It ends with consecutive Table 4 and Figures 8, 9 and 10. It should be done in an easier to understand form.

18.    The absence of a Discussion section is striking. This is unacceptable, and a Discussion section should be provided in which it is necessary to compare the results obtained with the results of other authors.

19.    The Conclusions are also rather succinct. Probably, it is necessary to record the obtained result in more detail and indicate the prospects for the development of the study in the conclusions.

Author Response

(The authors gave the same response as above.)

Reviewer 3 Report

The abstract is very verbally and do not have any evidence of using sensors as journal scope  !

What are the “large-size details” not clear what does means here details

Actually it is difficult to see authors contribution as there are referring numerous citation in material and methods !

The boundary condition and loading for the proposed model are none

Please check the entire paper for different typos “od the” – everywhere were found some issue

 A section of discussion is required as there is no a proper interpretation of results

As indicated in abstract the conclusion are very verbally they requires actually some quantitative data to be linked to your results

Most of the figure contain limited interpretation and very general details – please elaborate on them

Some more recent references are required

Author Response

(The authors gave the same response as above.)

Round 2

Reviewer 1 Report

Accept in present form.

Reviewer 2 Report

All my comments were taken into account and necessary corrections were made in the article's text. I recommend the article for publishing. 

Reviewer 3 Report

.